# Semantic-aware Pruning of Large Language Models via Neuron Importance Explanation

## Abstract

Large language models (LLMs) demonstrate unprecedented capabilities across diverse applications, yet their extensive parameterization creates substantial computational and memory requirements that hinder practical deployment. While structured pruning shows promise for LLM compression, existing methods use static masks that cannot adapt to different inputs, limiting performance across diverse tasks. In this work, we present SEAP, a novel semantic-aware structured pruning framework that adaptively identifies optimal masks based on input semantics at the pre-fill stage. Our framework features two key components: (1) an explainability-guided importance estimation that uniquely fuses local and global neuron importance to discover diverse representative mask patterns from calibration data's intrinsic characteristics, and (2) a lightweight router-based module through iterative refinement that efficiently assigns optimal masks for each input prompt. Experimental results on LLaMA-2/3, Qwen2, and Phi-2 demonstrate that SEAP outperforms state-of-the-art structured pruning methods across diverse language modeling and commonsense reasoning tasks, achieving competitive performance with reductions in memory and inference latency.

## 1 Introduction

Recent advancements in Large Language Models (LLMs) Liu et al. (2024); Achiam et al. (2023); Grattafiori et al. (2024) have demonstrated unprecedented capabilities across diverse applications Radford et al. (2021); Brown et al. (2020). However, their extensive parameterization imposes prohibitive computational and memory requirements, hindering deployment in resource-constrained environments Sun et al. (2024). Structured pruning Wen et al. (2016); An et al. (2024); Ashkboos et al. (2024) addresses this challenge by removing entire matrix components, achieving compression and acceleration while preserving hardware-friendly dense operations.

Most existing structured pruning methods adopt static approaches, deriving a single pruning pattern from calibration data using heuristic criteria An et al. (2024); Yang et al. (2023) or reconstruction losses van der Ouderaa et al. (2023); Guo et al. (2025). This *one-size-fits-all* paradigm applies identical patterns regardless of input semantics, which can lead to inconsistent performance across diverse downstream distributions and domain shifts Ji et al. (2025); Williams & Aletras (2024). In practice, a single globally optimal subnetwork may not exist: prompts with different linguistic styles, knowledge requirements, or reasoning structures often activate distinct functional pathways inside the same backbone.

Inspired by interpretability research on attribution circuits within LLMs Conmy et al. (2023); Merullo et al. (2024), we find that optimal computational pathways can vary systematically across semantic contexts. This motivates the development of semantic-aware pruning strategies that adapt sparsity patterns to input characteristics rather than applying static compression. Yet such an approach faces several challenges. First, pattern selection must occur before the pre-fill stage, without access to intermediate hidden states, precluding the use of input-specific importance metrics at inference time. Second, directly learning a sparsity predictor conditioned on input features is prohibitively expensive given the massive parameter scale of LLMs, often necessitating joint optimization of the model and predictor Hou et al. (2025). While recent work Wee et al. (2025) explores task-dependent pruning by selecting different transformer blocks for downstream tasks, such block-level adaptation provides limited flexibility compared to neuron-level pruning. These constraints

Figure 1: Motivation for semantic-aware pruning. Prompts with different semantics could activate different regions of LLMs. Static pruning applies a single one-size-fits-all mask that cannot accommodate such variability. SEAP discovers diverse structured patterns and dynamically routes each input to the suitable subnetwork, improving performance across tasks.

lead us to a key insight: while individual inputs exhibit unique activation patterns, there exist *shared* structured pruning patterns across semantically similar prompts that can be discovered once and then reused efficiently via lightweight routing.

To achieve this, we introduce SEAP, a semantic-aware *dynamic* structured pruning framework that learns a small pool of specialized sparse subnetworks from calibration data and dynamically assigns an appropriate configuration to each input during inference. Figure 1 illustrates this shift from static uniform patterns to semantic-aware mask selection. Crucially, SEAP keeps the LLM backbone *frozen* and learns only a lightweight BERT-based router that selects among a small pool of representative subnetworks. Especially, we propose an explainability-guided importance estimation method that leverages neuron attribution techniques Achtibat et al. (2024); Ali et al. (2022) to bridge the gap between individual neuron relevance and combinatorial pruning decisions. By fusing local activations with global relevance signals, we discover how different semantic contexts engage distinct computational pathways, enabling the extraction of multiple effective structured pruning patterns from calibration data. SEAP achieves semantic adaptivity while maintaining efficiency through a two-phase design. We first distill the diverse learned patterns into a compact candidate pool via maximum-coverage optimization, balancing representativeness with efficiency. At the pre-fill stage, a lightweight router encodes input semantics and selects the optimal pattern. To jointly optimize mask assignment and specialization, we employ iterative co-training where the router learns to predict mask performance while individual masks adapt to their assigned contexts. This alternating optimization yields complementary subnetworks that align with input semantics.

Extensive experimental results show that our method can outperform state-of-the-art structural pruning methods for LLMs while still maintaining low computational costs. Our main contributions can be summarized as follows:

- We propose SEAP, a novel dynamic structured pruning framework that explores the intrinsic characteristics of calibration data for semantic-aware pruning.
- We bridge neuron attribution and structured pruning by designing an explainability-guided importance estimation method that discovers diverse mask patterns.
- Extensive experiments on LLaMA-2/3, Qwen2, and Phi-2 demonstrate that SEAP consistently outperforms state-of-the-art structured pruning methods across diverse language modeling and reasoning tasks while maintaining hardware efficiency.

## 2 RELATED WORK

Current pruning approaches fall into three main categories, each with distinct characteristics and applications. Unstructured pruning removes individual weights to create irregular sparsity patterns, achieving high compression ratios Frantar & Alistarh (2023); Sun et al. (2024). Semi-structured pruning Fang et al. (2024); Zhang et al. imposes fine-grained N:M sparsity patterns (e.g., 2:4) that can leverage specialized hardware accelerators for efficient execution. Structured pruning Xia et al. (2024); Ma et al. (2023) removes entire components such as neurons, attention heads, or blocks, enabling straightforward deployment on standard hardware through dense matrix operations.

Within structured pruning, representative methods include heuristic scoring based on weight magnitudes, gradients, or similarity metrics An et al. (2024); Kurtic et al. (2023); Ma et al. (2023); Guo et al. (2025); Chen et al. (2025a), matrix-decomposition-based approaches such as SliceGPT and SVD-LLM Ashkboos et al. (2024); Wang et al. (2025), and optimization-based frameworks like DISP-LLM Gao et al. (2024). Despite their algorithmic differences, these methods ultimately learn a *static* structured pruning pattern from calibration data and apply it uniformly to all inputs, assuming a single subnetwork can serve diverse semantic contexts.

Complementary to static pruning, contextual sparsity methods Liu et al. (2023b); Zhou et al. (2024); Lee et al. (2024) explore dynamic component activation during inference, focusing on token-level acceleration while preserving full model parameters. Recent dynamic approaches Wee et al. (2025); Hou et al. (2025) have begun exploring input-adaptive pruning in the pre-fill stage. For example, IFPruning Hou et al. (2025) jointly trains a sparsity predictor with the LLM to generate input-specific pruning masks, while Pudding Wee et al. (2025) selects different transformer blocks for various downstream tasks. These methods tightly couple the sparsity predictor with backbone optimization, or operate at coarse block granularity, which can limit flexibility or increase training complexity.

SEAP extends structured pruning by introducing semantic adaptivity through multiple specialized sparse subnetworks. Our approach combines the memory efficiency of structured pruning with input-aware adaptation: we select appropriate sub-models during pre-fill and maintain consistent parameters throughout decoding, avoiding per-token routing overhead. By operating at fine-grained neuron granularity guided by explainability signals, SEAP focuses on semantic-aware structured pruning within a significantly expanded mask search space.

## 3 PRELIMINARIES

### 3.1 OVERVIEW OF STRUCTURED PRUNING

LLMs are composed of $L$ Transformer blocks, each containing a multi-head self-attention mechanism $\text{MHA}(\cdot)$ and a feed-forward network $\text{FFN}(\cdot)$. Denote the input hidden state $\mathbf{X}_{\text{in}} \in \mathbb{R}^{n \times d}$, where $n$ and $d$ represent the sequence length and hidden dimension, respectively. Considering the residual connections, the transformations of each block can be expressed as:

$$\mathbf{X}_{\text{res}} = \text{MHA}(\mathbf{X}_{\text{in}}) + \mathbf{X}_{\text{in}}, \qquad \mathbf{X}_{\text{out}} = \text{FFN}(\mathbf{X}_{\text{res}}) + \mathbf{X}_{\text{res}}. \tag{1}$$

Commonly, the attention layer has four matrices: $\mathbf{W}_q, \mathbf{W}_k, \mathbf{W}_v, \mathbf{W}_o \in \mathbb{R}^{d \times d}$, and the feed-forward layer has three matrices: $\mathbf{W}_{\text{up}}, \mathbf{W}_{\text{gate}} \in \mathbb{R}^{d \times d_{\text{ffn}}}, \mathbf{W}_{\text{down}} \in \mathbb{R}^{d_{\text{ffn}} \times d}$, where $d_{\text{ffn}}$ is the hidden dimension.

Without loss of generality, we adopt the recently proposed dimension-independent pruning (DIP) framework Gao et al. (2024) for structured pruning, which allows us to prune each layer independently through indexing operations. Our objective is to identify a set of pseudo-index selection matrices $\{\mathbf{S}_i\}_{i=1}^5$ in each transformer block. $\mathbf{S}_i$ is defined as a diagonal binary matrix and the position of the ones indicating the selection of specific neurons. Based on these selection matrices, the attention and feed-forward layers can be expressed as:

$$\text{MHA}(\mathbf{X}) = \text{SDPA}(\mathbf{X}\mathbf{S}_1^T\mathbf{W}_q, \mathbf{X}\mathbf{S}_1^T\mathbf{W}_k, \mathbf{X}\mathbf{S}_1^T\mathbf{W}_v)\mathbf{W}_o\mathbf{S}_2,$$
$$\text{FFN}(\mathbf{X}) = \left(\sigma(\mathbf{X}\mathbf{S}_3^T\mathbf{W}_{\text{up}}\mathbf{S}_4) \odot (\mathbf{X}\mathbf{S}_3^T\mathbf{W}_{\text{gate}}\mathbf{S}_4)\right)\mathbf{S}_4^T\mathbf{W}_{\text{down}}\mathbf{S}_5, \tag{2}$$

where $\text{SDPA}(\cdot)$ is the scaled dot-product attention kernel and $\sigma(\cdot)$ is the element-wise activation function. While we present the formulation under DIP, this selection-matrix approach can generalize to other structured pruning schemes Ma et al. (2023); An et al. (2024).

### 3.2 LAYER-WISE RELEVANCE PROPAGATION

Layer-wise Relevance Propagation (LRP) Bach et al. (2015) is a neuron-attribution algorithm that redistributes a model's output score back through the network, layer by layer. The relevance value $R_j^{(l)}$ quantifies the contribution of neuron $j$ in layer $l$ to the final prediction. For linear transformations, the propagation rule is

$$R_i^{(l-1)} = \sum_j \frac{a_i^{(l-1)} w_{ij}^{(l)}}{\sum_k a_k^{(l-1)} w_{kj}^{(l)} + \epsilon} R_j^{(l)}, \tag{3}$$

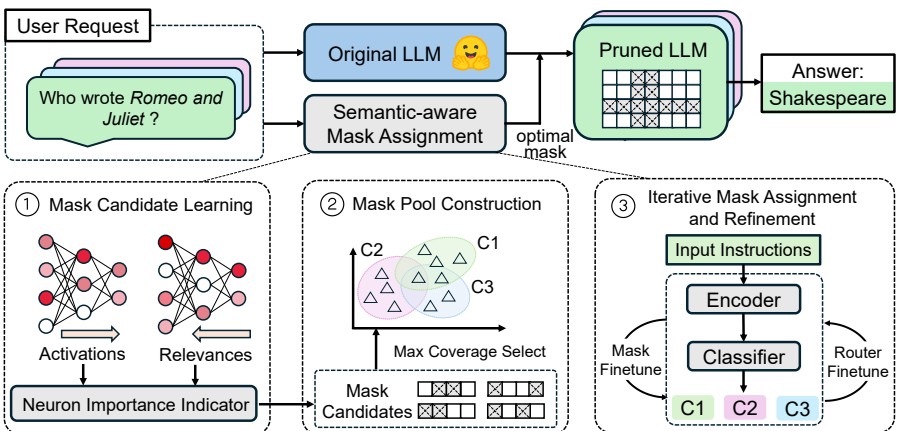

Figure 2: Overview of SEAP's semantic-aware pruning framework: (i) learning diverse mask candidates from calibration data, (ii) constructing a compact candidate pool, and (iii) dynamically selecting masks based on input semantics.

where $a_i^{(l-1)}$ is the activation of neuron $i$ in layer $l-1$, $w_{ij}^{(l)}$ is the weight from neuron $i$ to neuron $j$, and $\epsilon$ is a small stabilization constant. In the transformer context, recent studies Achtibat et al. (2024); Ali et al. (2022) extend this rule to non-linear components (e.g., attention) through local linearization via Deep Taylor Decomposition Sixt & Landgraf (2022). Formally, LRP maintains a conservation principle: $\sum_i R_i^{(l-1)} = \sum_j R_j^{(l)}$, ensuring constant total relevance flow between adjacent layers. Unlike activation values or gradients, LRP scores capture global network computations beyond local property, making them well-suited for guiding structured pruning.

## 4 METHODOLOGY

The objective of SEAP is to dynamically learn optimal pruning masks based on input semantics. As shown in Figure 2, SEAP consists of three main stages: (i) **mask candidate learning** (Section 4.1), which learns instance-specific pruning masks for each calibration sample using explainability-guided neuron importance estimation; (ii) **mask pool construction** (Section 4.2), which distills the learned masks into a compact set of representative and diverse candidates; and (iii) **router-based mask assignment** (Section 4.3), which trains a lightweight module to dynamically select and optimize the best mask from the candidate pool for each input.

### 4.1 MASK CANDIDATE LEARNING

To identify optimal structured pruning patterns, we exploit calibration data $\mathcal{D}_{\text{cal}}$ whose distribution approximates that of pre-training data. While different inputs may require different pruning strategies, we find that shared pruning patterns could emerge across semantically similar inputs. This key insight suggests that a compact set of representative masks could efficiently capture the pruning requirements of diverse inputs, motivating our approach to learn instance-specific masks and subsequently distill them into a reusable candidate pool.

For a fixed LLM with parameters $\mathbf{W}$ consisting of $L$ blocks, we learn to map each input sequence $\mathbf{X} \in \mathbb{R}^{n \times d}$ (where $n$ is the sequence length and $d$ is the hidden dimension) to a binary selection matrix $\mathbf{S}^{(l)}(\mathbf{X}) \in \mathbb{R}^{d \times d}$ for each layer $l$ that determines which neurons to retain. Here, we omit component-wise notation for clarity. To enable adaptive sparsity allocation across layers, we formulate instance-specific mask learning as a global optimization problem over the calibration set:

$$\min_{\Theta} \sum_{\mathbf{X} \in \mathcal{D}_{\text{cal}}} \left[ \mathcal{L}(\mathbf{X}; \mathbf{W}, \mathbf{S}(\mathbf{X})) + \lambda \mathcal{R}(\mathbf{S}(\mathbf{X}), p) \right], \tag{4}$$

where $\mathcal{L}(\mathbf{X}; \mathbf{W}, \mathbf{S}(\mathbf{X}))$ denotes the language modeling loss computed on the pruned network, $\mathcal{R}(\mathbf{S}(\mathbf{X}), p)$ enforces the target sparsity constraint $p$, $\lambda$ is the regularization weight, and $\Theta$ denotes all learnable parameters in the mask learning network.

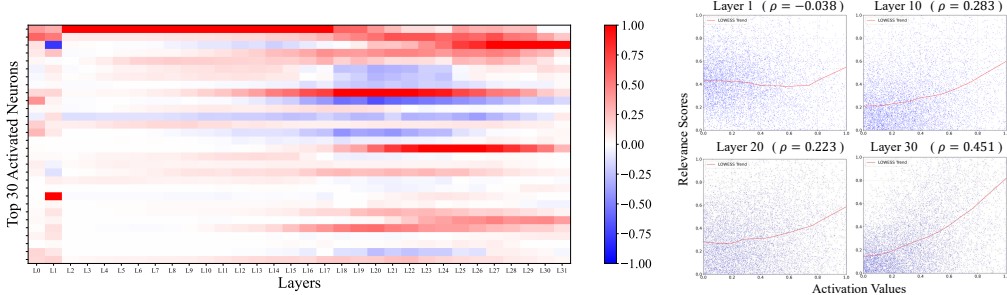

Figure 3: Heatmap of LRP scores for top-activated neurons (0.7% per layer) across layers. Red/blue indicate positive/negative output impact; color intensity shows its magnitude. Right: LRP-activation scatter plots for selected layers with LOWESS trend lines (red), showing correlation coefficients ($\rho$).

**Explainability-Guided Neuron Importance Indicator.** Traditional magnitude-based criteria (e.g., activations) are insufficient for reliable neuron importance estimation, as they capture only local information while neglecting critical cross-layer interdependencies. As illustrated in Figure 3, many highly-activated neurons exhibit low relevance scores, particularly in early layers, demonstrating that activation magnitude poorly correlates with prediction contribution. This disconnect is quantified by varying Spearman correlations across layers ($\rho \in [-0.038, 0.451]$), revealing a complex, non-monotonic relationship. Moreover, LOWESS analysis shows that magnitude-importance relationships become unpredictable in low-activation regions, indicating systematic bias in activation-based selection. These findings motivate our integration of complementary explainability signals for comprehensive importance assessment.

For each layer $l$ with input hidden states $\mathbf{X}^{(l)} \in \mathbb{R}^{n \times d}$, we compute two complementary per-neuron importance signals to capture both local activation patterns and global prediction relevance:

$$\mathbf{A}^{(l)} = \left[ \left\| \mathbf{X}^{(l)}_{:,1} \right\|_2, \left\| \mathbf{X}^{(l)}_{:,2} \right\|_2, \ldots, \left\| \mathbf{X}^{(l)}_{:,d} \right\|_2 \right]^T \in \mathbb{R}^d, \tag{5}$$

$$\mathbf{R}^{(l)} = \text{LRP-Rule}\left( \mathbf{R}^{(l+1)}, \mathbf{W}^{(l \to l+1)}, \mathbf{X}^{(l)} \right) \in \mathbb{R}^d, \tag{6}$$

where $\mathbf{A}^{(l)}$ aggregates L2 norms across sequence positions for each dimension, and LRP-Rule propagates relevance backward from layer $l+1$ using decomposition rules Achtibat et al. (2024), with output layer relevance initialized based on the language modeling loss.

To quantify neuron importance, we combine these signals into a unified indicator via a lightweight channel-wise MLP that balances local and global perspectives:

$$\mathbf{I}^{(l)} = \alpha |\mathbf{R}^{(l)}| + (1 - \alpha) \text{MLP}^{(l)}(\mathbf{A}^{(l)}), \quad \alpha \in [0, 1], \quad \text{MLP}^{(l)} : \mathbb{R}^d \to \mathbb{R}^d. \tag{7}$$

This convex combination allows the MLP to capture input-specific activation variations, while the weighted relevance term $\alpha |\mathbf{R}^{(l)}|$ provides global contribution context.

Recognizing that uniform sparsity across layers is suboptimal, we introduce per-layer affine transformations to automatically adjust importance distributions while preserving relative neuron rankings. Specifically, SEAP learns per-layer scaling factors and biases that adjust the importance scores:

$$\hat{\mathbf{I}}^{(l)} = \gamma^{(l)} \mathbf{I}^{(l)} + \beta^{(l)}, \quad \gamma^{(l)} > 0, \quad \beta^{(l)} \in \mathbb{R}, \tag{8}$$

where $\gamma^{(l)}$ adjusts the dynamic range and $\beta^{(l)}$ aligns the distribution with pruning thresholds.

Finally, we generate structured selection matrices by applying the recently proposed gradient estimator Binary ReinMax Liu et al. (2023a) to the neuron-level importance scores:

$$\mathbf{m}^{(l)}(\mathbf{X}) = \text{ReinMax}(\hat{\mathbf{I}}^{(l)}), \quad \mathbf{S}^{(l)}(\mathbf{X}) = \text{Diag}(\mathbf{m}^{(l)}(\mathbf{X})) \in \mathbb{R}^{d \times d}, \tag{9}$$

where $\mathbf{m}^{(l)}(\mathbf{X}) \in \{0, 1\}^d$ is the binary mask and $\mathbf{S}^{(l)}(\mathbf{X})$ is the corresponding diagonal selection matrix. ReinMax maintains discrete forward computation while enabling end-to-end differentiable optimization of $\Theta = \{\alpha, \{\text{MLP}^{(l)}, \gamma^{(l)}, \beta^{(l)}\}_{l=1}^L\}$ to optimize sparsity-performance trade-offs.

### 4.1.1 Mask Pool Construction

While our mask candidate learning process generates a diverse collection of masks $\mathcal{U}$ from calibration data, deploying all masks at inference would introduce prohibitive computational overhead. Therefore, we distill these masks into a compact representative candidate set $\mathcal{M}$ that retains pruning diversity while maintaining efficiency.

We begin by evaluating each learned mask candidate $\mathbf{m}$ on a stratified subset randomly sampled from downstream task distributions. Let $\mathcal{S}(\mathbf{m})$ denote the set of validation instances where mask $\mathbf{m}$ achieves correct predictions. The mask selection problem can be formulated as a maximum coverage optimization problem:

$$\mathcal{M}^* = \underset{\mathcal{M} \subseteq \mathcal{U}:|\mathcal{M}| \leq K}{\arg\max} \left| \bigcup_{\mathbf{m} \in \mathcal{M}} \mathcal{S}(\mathbf{m}) \right|, \tag{10}$$

where $K$ is the maximum number of candidate masks. To solve this efficiently, we employ a greedy algorithm: starting with an empty set, we iteratively add the mask that provides the highest incremental coverage until reaching the limit $K$.

As illustrated in Figure 4, our experiments on LLaMA-2-7B reveals that most learned masks in SEAP achieve higher accuracy compared to the static baseline Gao et al. (2024). Notably, combining 16 candidate masks achieves 87% coverage of the validation set, around 57.5% improvement over static baseline. This evidence demonstrates that different masks capture complementary pruning patterns, motivating our instance-adaptive mask assignment approach.

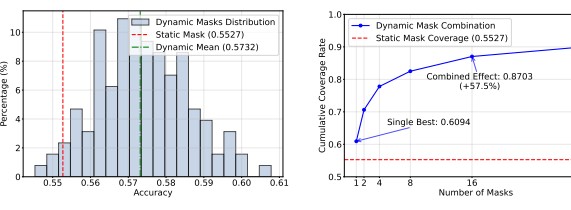

(a) Single Mask Accuracy.    (b) Combined Mask Coverage.

Figure 4: Accuracy distribution and cumulative coverage of learned mask candidates on LLaMA-2-7B at 40% ratio.

## 4.2 Router-based Mask Assignment and Refinement

Given the candidate mask pool $\mathcal{M}$, we formulate instance-adaptive mask selection as a classification problem where a lightweight router predicts the optimal mask for each input. Our router architecture comprises: (1) a much smaller LLM backbone ($\sim$150M parameters) for semantic encoding, and (2) a two-layer MLP classifier that maps encoded representations to mask selections from $\mathcal{M}$.

To train the router, we construct a supervision dataset capturing the relationship between input characteristics and optimal mask performance. For multiple-choice tasks, each training sample consists of a question $\mathbf{X}$, candidate options $\mathcal{A} = \{a_1, \ldots, a_J\}$, and ground-truth option index $j^*$. We evaluate mask quality using an option-wise discriminative score that measures how well the pruned model $\mathbf{W} \odot \mathbf{m}_c$ (where $\odot$ denotes element-wise masking) distinguishes correct answers from distractors:

$$s^{(c)} = -\log \frac{\exp\left(-\mathrm{tl}(\mathbf{X}, a_{j^*}; \mathbf{W} \odot \mathbf{m}_c)\right)}{\sum_{j=1}^{J} \exp\left(-\mathrm{tl}(\mathbf{X}, a_j; \mathbf{W} \odot \mathbf{m}_c)\right)}, \tag{11}$$

where $\mathrm{tl}(\mathbf{X}, a_j; \mathbf{W} \odot \mathbf{m}_c)$ denotes the negative log-likelihood of generating option $a_j$, and $s^{(c)}$ represents the negative log-probability of mask $\mathbf{m}_c$ correctly predicting $j^*$.

We employ iterative co-optimization to jointly refine router predictions and mask specialization. In each iteration, we first update the router using LoRA fine-tuning with MSE regression on per-mask performance scores:

$$\mathcal{L}_{\text{router}} = \frac{1}{N} \sum_{i=1}^{N} \|\mathbf{s}_i - \hat{\mathbf{s}}_i\|_2^2, \quad \mathbf{s}_i = [s_i^{(1)}, s_i^{(2)}, \ldots, s_i^{(K)}], \tag{12}$$

where $s_i^{(c)}$ is computed via Equation 11 for sample $i$ under mask $\mathbf{m}_c$, and $\hat{\mathbf{s}}_i$ represents the router's prediction. Subsequently, we fix router parameters and update mask scores using the refined predictions as supervision. Through this alternating optimization, the router learns to associate input semantics with effective pruning patterns, while masks adapt to their assigned semantic contexts.

## 4.3 COMPUTATIONAL COMPLEXITY

SEAP keeps the backbone LLM frozen and only optimizes lightweight auxiliary modules. Let $N_{\text{cal}}$ be the number of calibration examples, $K$ the number of candidate masks, and $L$ the number of layers. The mask-generation stage has cost $\mathcal{O}\big(N_{\text{cal}} \cdot K \cdot L\big)$, with gradients flowing through the hypernetwork but not the backbone. Router training operates on a compact encoder (e.g., Modern-Bert Warner et al. (2025)) of size $|\theta_{\text{router}}| \approx 150M \ll |\theta_{\text{LLM}}|$ with parameter-efficient adaptation, so its complexity scales as $\mathcal{O}\big(N_{\text{train}} \cdot |\theta_{\text{router}}|\big)$, largely decoupled from the backbone size.

At inference time, for each input sequence we perform one router forward and one subnetwork reconstruction, both of cost $\mathcal{O}(|\theta_{\text{router}}|)$, after which the LLM forward FLOPs match those of a static structured-pruned model with sparsity ratio $s$ (roughly $(1-s)$ of the dense FLOPs). Memory usage is dominated by a single dense backbone plus $\mathcal{O}(|\theta_{\text{router}}| + K \cdot N_{\text{neurons}})$ for the router and binary masks, corresponding to only a small ($\sim$2–3%) overhead in practice. Empirical training-time, latency, and transfer-cost measurements in Tables 4 to 6 further confirm that this additional cost remains modest in both server-side and memory-constrained deployments.

## 5 EXPERIMENTS

In this section, we present an experimental evaluation of SEAP, covering the setup, results on language modeling and reasoning/QA benchmarks, and analyses of routing behavior and efficiency.

### 5.1 EXPERIMENTAL SETUP

**Models, Datasets, and Baselines** We primarily evaluate SEAP on LLaMA-2-7B, LLaMA-2-13B, and Phi-2, and further validate its scalability and robustness on Qwen-2-7B and LLaMA-3-8B. For language modeling, we report perplexity (PPL) on WikiText-2 Merity et al. (2017) and PTB Marcus et al. (1993). For downstream evaluation, we follow the official implementation of LM-Eval-Harness Gao et al. (2023) and use a suite of zero-shot commonsense and knowledge reasoning tasks, including ARC-Easy Clark et al. (2018), ARC-Challenge Clark et al. (2018), Winogrande Sakaguchi et al. (2020), HellaSwag Zellers et al. (2019) and PIQA Bisk et al. (2020). To assess the robustness of the semantic router under calibration distribution shifts, we additionally evaluate on BoolQ Clark et al. (2019), OpenBookQA (OBQA) Mihaylov et al. (2018), MBPP Austin et al. (2021), PubMedQA Jin et al. (2019), MMLU Hendrycks et al. (2021), and SciQ Welbl et al. (2017), as presented in Table 2 and Table 3. We compare SEAP with SOTA structured pruning methods that operate at neuron or channel granularity while preserving dense kernels. These include FLAP An et al. (2024), SliceGPT Ashkboos et al. (2024), ShortGPT Men et al. (2025), and DISP-LLM Gao et al. (2024).

**Implementation Details** For mask candidate generation, we use 128 randomly sampled 2048-token chunks from the WikiText-2 and Alpaca Taori et al. (2023) training sets as calibration data. We evaluate learned mask candidates on 128 samples from the training split of five commonsense reasoning datasets (ARC-Easy, ARC-Challenge, Winogrande, HellaSwag, and PIQA), and select the top-10 mask candidates for inference by default. For router training, we construct the training set using the full training splits of the same five datasets, while reserving BoolQ, OBQA, MBPP, PubMedQA, MMLU, and SciQ exclusively for evaluation to assess generalization to unseen tasks. Unless otherwise specified, all training and analysis are conducted on a server with 8 NVIDIA A6000 GPUs. Additional implementation details are provided in the Appendix.

### 5.2 OVERALL PERFORMANCE

**Language Modeling** We first evaluate the perplexity of compressed models on WikiText-2 and PTB, with results reported in Table 1. Across both model sizes and compression ratios, SEAP consistently achieves the lowest perplexity among all structured pruning baselines. At 20% compression on LLaMA-2-7B, SEAP improves over the second-best method DISP-LLM by 12.5% on WikiText-2 and 12.8% on PTB. The gap widens at 40% compression, where alternative methods suffer severe degradation (e.g. PTB), while SEAP maintains significantly better perplexity. Similar trends hold for LLaMA-2-13B, indicating that semantic-aware routing not only preserves downstream accuracy but also leads to more faithful language modeling under aggressive structured pruning.

**Commonsense Reasoning** As shown in Table 2, SEAP better preserves zero-shot commonsense and knowledge reasoning capabilities on LLaMA-2-7B, LLaMA-2-13B, and Phi-2. At both 20% and 40% compression ratios, SEAP consistently achieves the highest average accuracy across BoolQ, PIQA, HellaSwag, OBQA, ARC-e, ARC-c, and WinoGrande, and remains substantially closer to the dense models than FLAP, SliceGPT, ShortGPT, and DISP-LLM. This pattern holds even at higher sparsity, where static structured methods exhibit pronounced degradation.

Table 1: Perplexity of the compressed LLaMA-2-7B and LLaMA-2-13B models on WikiText-2 and PTB.

| Ratios | Methods | LLaMA-2-7B | | LLaMA-2-13B | |
|---|---|---|---|---|---|
| | | WikiText | PTB | WikiText | PTB |
| 0% | Dense | 5.47 | 24.09 | 4.88 | 35.03 |
| 20% | FLAP | 16.33 | 67.58 | 14.74 | 85.86 |
| | ShortGPT | 15.76 | 100.06 | 8.32 | 132.92 |
| | SliceGPT | 8.12 | 84.14 | 7.16 | 92.62 |
| | DISP-LLM | 6.98 | 53.89 | 5.98 | 87.31 |
| | Ours | **6.11** | **46.97** | **5.37** | **55.96** |
| 40% | FLAP | 28.12 | 112.06 | 21.51 | 150.11 |
| | ShortGPT | 75.22 | 260.58 | 58.22 | 332.74 |
| | SliceGPT | 14.27 | 194.69 | 12.24 | 252.98 |
| | DISP-LLM | 10.34 | 165.67 | 8.42 | 205.99 |
| | Ours | **8.52** | **86.48** | **7.86** | **112.05** |

The results on Qwen-2-7B and LLaMA-3-8B in Table 3 are consistent with this trend. At 25% sparsity, SEAP generally outperforms DISP-LLM on the six reasoning benchmarks while also achieving lower perplexity on WikiText-2. On the OOD suite (BoolQ, OBQA, MBPP, PubMedQA, MMLU, SciQ), SEAP recovers a larger fraction of dense performance than DISP-LLM in most cases, and even slightly exceeds the dense model on some datasets (e.g., SciQ with Qwen-2-7B). These observations suggest that semantic routing can remain effective beyond the calibration distribution, including on code generation and domain-shifted science/knowledge tasks. We believe this benefit comes from the good quality of the learned subnetworks and the fine-tuned router, which together preserves input semantic adaptivity.

## 5.3 IN-DEPTH ANALYSIS

**Impact of Number of Candidate Masks** Figure 5a illustrates the performance of our method with varying numbers of candidate masks $K$. We can observe that at the initial stage, increasing the number of candidate masks leads to a noticeable improvement in performance. While $K$ equals 1, the model is essentially static pruning. This shows the necessity of a diverse set of mask patterns to capture different input characteristics and enhance the model's reasoning capabilities. However, as $K$ continues to increase to 20, the performance gains become marginal or even negative. This indicates a clear trade-off: a modest pool of candidate masks is essential, but excessive diversity offers little benefit and may introduce unnecessary stochasticity.

**Ablation Study** As presented in Figure 5b, we conduct an ablation study to evaluate the effectiveness of each component in our method. Replacing the option-wise log-softmax loss with task-likelihood loss yields the most significant performance drop, demonstrating that it enables more discriminative mask assignments by learning nuanced input space boundaries rather than optimizing solely for task-specific performance. Removing the mask

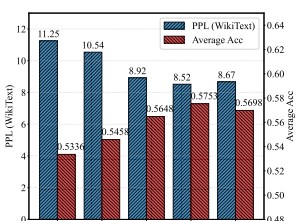
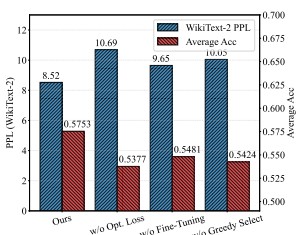

(a) Impact of candidate number  (b) Component ablation study

Figure 5: Sensitivity and ablation of SEAP. (a) Performance with varying numbers of masks $K$. (b) Ablation effects.

fine-tuning step results in moderate degradation, indicating that this calibration mechanism is essential for adapting learned masks to compressed model dynamics while preserving routing logic. Moreover, substituting greedy mask selection with random sampling leads to substantial performance deterioration, highlighting that systematic input space coverage is crucial for robust router training and generalization to unseen inputs.

**Adaptive Layer-wise Importance Discovery** One of the advantages of our method is that it learns a global importance criterion for each structure, allowing to pruning models without the need for layer-wise pruning ratios. As shown in Figure 6, we visualize the averaged pruning ratios of learned

Table 2: Downstream task accuracy of the compressed LLaMA-2-7B, LLaMA-2-13B, and Phi-2 models. **Bold** denotes the best result at the same compression ratio.

| Model | Pruning Ratio | Method | BoolQ acc | PIQA acc_norm | HellaSwag acc_norm | OBQA acc_norm | ARC-e acc_norm | ARC-c acc_norm | WinoGrande acc | Average |
|---|---|---|---|---|---|---|---|---|---|---|
| LLaMA-2-7B | 0% | Raw | 0.7798 | 0.7889 | 0.7618 | 0.4480 | 0.7416 | 0.4565 | 0.6938 | 0.6672 |
| | 20% | FLAP | 0.6168 | 0.7323 | 0.6371 | 0.3760 | 0.6199 | 0.3729 | 0.6346 | 0.5699 |
| | | ShortGPT | 0.5217 | 0.6012 | 0.4343 | 0.2900 | 0.5227 | 0.3208 | 0.5808 | 0.4674 |
| | | SliceGPT | 0.5196 | 0.6425 | 0.4978 | 0.2900 | 0.5147 | 0.3106 | 0.6274 | 0.4861 |
| | | DISP-LLM | 0.6774 | 0.7399 | 0.6776 | 0.3780 | 0.6595 | 0.3686 | 0.6440 | 0.5921 |
| | | **SeAP** | **0.7370** | **0.7652** | **0.7128** | **0.4020** | **0.6897** | **0.4051** | **0.6677** | **0.6256** |
| | 40% | FLAP | 0.5547 | 0.6605 | 0.4734 | 0.3460 | 0.3809 | 0.2816 | 0.5501 | 0.4639 |
| | | ShortGPT | 0.4557 | 0.5098 | 0.2783 | 0.2660 | 0.2639 | 0.2747 | 0.5051 | 0.3648 |
| | | SliceGPT | 0.4713 | 0.5490 | 0.3480 | 0.2540 | 0.3068 | 0.2346 | 0.4949 | 0.3798 |
| | | DISP-LLM | 0.6361 | 0.7084 | 0.5482 | 0.3620 | 0.5379 | 0.3208 | 0.5730 | 0.5266 |
| | | **SeAP** | **0.6869** | **0.7367** | **0.5929** | **0.3880** | **0.6147** | **0.3582** | **0.6495** | **0.5753** |
| LLaMA-2-13B | 0% | Raw | 0.8070 | 0.8047 | 0.7937 | 0.4560 | 0.7761 | 0.4940 | 0.7182 | 0.6928 |
| | 20% | FLAP | 0.6239 | 0.7546 | 0.6886 | 0.3940 | 0.6587 | 0.4070 | 0.6527 | 0.5971 |
| | | SliceGPT | 0.5776 | 0.6583 | 0.5358 | 0.3220 | 0.5581 | 0.3584 | 0.6717 | 0.5260 |
| | | DISP-LLM | 0.7416 | 0.7666 | 0.7366 | 0.4220 | 0.7281 | 0.4437 | 0.6677 | 0.6438 |
| | | **SeAP** | **0.7611** | **0.7797** | **0.7512** | **0.4340** | **0.7507** | **0.4763** | **0.7001** | **0.6647** |
| | 40% | FLAP | 0.6245 | 0.6866 | 0.5397 | 0.3540 | 0.5311 | 0.3481 | 0.6117 | 0.5280 |
| | | SliceGPT | 0.4723 | 0.5598 | 0.3715 | 0.2960 | 0.3859 | 0.2705 | 0.5651 | 0.4173 |
| | | DISP-LLM | 0.6547 | 0.6986 | 0.5967 | 0.3680 | 0.5909 | 0.3396 | 0.5801 | 0.5469 |
| | | **SeAP** | **0.7297** | **0.7245** | **0.6240** | **0.3860** | **0.6693** | **0.3692** | **0.6653** | **0.5954** |
| Phi-2 | 0% | Raw | 0.8226 | 0.8123 | 0.7889 | 0.4460 | 0.8098 | 0.5358 | 0.7364 | 0.7074 |
| | 20% | SliceGPT | 0.7269 | 0.7187 | 0.5776 | 0.3740 | 0.5800 | 0.3532 | 0.6780 | 0.5726 |
| | | DISP-LLM | 0.7318 | 0.7486 | 0.6293 | 0.3740 | 0.6818 | 0.4411 | 0.6709 | 0.6111 |
| | | **SeAP** | **0.7593** | **0.7679** | **0.6721** | **0.4120** | **0.7284** | **0.4735** | **0.6828** | **0.6423** |
| | 25% | SliceGPT | 0.6786 | 0.6991 | 0.5248 | 0.3460 | 0.5278 | 0.3549 | 0.6519 | 0.5404 |
| | | DISP-LLM | 0.7146 | 0.7427 | 0.5995 | 0.3520 | 0.6593 | 0.4334 | 0.6511 | 0.5932 |
| | | **SeAP** | **0.7483** | **0.7528** | **0.6465** | **0.3880** | **0.6684** | **0.4516** | **0.6779** | **0.6191** |
| | 30% | SliceGPT | 0.6478 | 0.6594 | 0.4756 | 0.3420 | 0.5303 | 0.3029 | 0.6314 | 0.5128 |
| | | DISP-LLM | 0.6989 | 0.7334 | 0.5443 | 0.3440 | 0.6359 | 0.3848 | 0.6322 | 0.5569 |
| | | **SeAP** | **0.7339** | **0.7476** | **0.6381** | **0.3760** | **0.6373** | **0.4327** | **0.6700** | **0.5943** |

Table 3: Downstream task accuracy of the compressed Qwen-2-7B and LLaMA-3-8B models at 25% sparsity. **Bold** denotes the best result at the same compression ratio.

| Model | BoolQ acc | PIQA acc_norm | HellaSwag acc_norm | OBQA acc_norm | ARC-e acc_norm | ARC-c acc_norm | WikiText2 PPL ↓ | MBPP pass@1 | PubMedQA acc | MMLU acc | SciQ acc_norm |
|---|---|---|---|---|---|---|---|---|---|---|---|
| **Qwen-2-7B** | | | | | | | | | | | |
| Dense | 0.8544 | 0.8063 | 0.8069 | 0.4640 | 0.7647 | 0.5392 | 7.60 | 0.5620 | 0.7440 | 0.6993 | 0.9150 |
| DISP-LLM | 0.7621 | 0.7693 | 0.6842 | 0.3880 | 0.7104 | 0.4556 | 11.62 | 0.2800 | 0.6040 | 0.5033 | 0.9130 |
| SeAP | **0.8263** | **0.7720** | **0.7351** | **0.4140** | **0.7357** | **0.5094** | **8.98** | **0.3820** | **0.6900** | **0.5801** | **0.9250** |
| **LLaMA-3-8B** | | | | | | | | | | | |
| Dense | 0.8226 | 0.8123 | 0.7889 | 0.4460 | 0.8098 | 0.5358 | 6.23 | 0.4820 | 0.7560 | 0.6338 | 0.9450 |
| DISP-LLM | 0.7214 | 0.7443 | 0.6738 | 0.4120 | 0.6965 | 0.4403 | 14.05 | 0.2020 | 0.5960 | 0.4133 | **0.8840** |
| SeAP | **0.7743** | **0.7639** | **0.6853** | **0.4210** | **0.7382** | **0.4629** | **11.20** | **0.2160** | **0.7100** | **0.4554** | 0.8770 |

candidate masks.The results demonstrate our method's understanding of transformer architecture functionality. Notably, middle layers exhibit lower pruning ratios compared to shallow and deep layers, aligning with recent findings that intermediate representations achieve superior performance across diverse tasks Skean et al. (2025). This pattern reflects the critical role of mid-depth layers in balancing information compression and semantic abstraction Chen et al. (2025b), indicating that our method effectively captures the nuanced dynamics of LLMs.

**Visualization of per-dataset Mask Assignment** Figure 7 visualizes the mask candidates selected for each dataset, revealing distinct selection patterns that validate our method's adaptability. Each task exhibits unique preferences: BoolQ strongly favors masks 6-9, while OpenBookQA predominantly selects masks 2-5, and PIQA shows concentrated usage of masks 5 and 6. These varied distributions demonstrate that our method effectively captures the nuanced characteristics of different datasets, allowing the model to dynamically adjust its routing logic based on input context. This adaptability is crucial for maintaining high performance across diverse tasks, as it enables the model to leverage the most relevant mask patterns for each specific reasoning challenge.

**Computational Cost Analysis** While SEAP employs a two-stage training procedure, the computational overhead remains practical and is well-justified by the performance gains. As detailed in Table 5, our training comprises explainability-guided mask candidate learning (1.56–2.56 hours) and dynamic router training (4.53–8.47 hours). Both stages operate on a *frozen* backbone: the mask

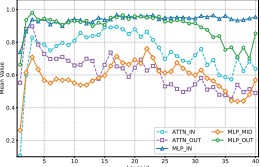
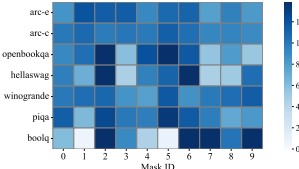

| Ratios | Mask Assignment | Parameter Loading | Pre-fill | TTFT | Decode |
|---|---|---|---|---|---|
| 0% | - | - | 0.704 | 0.704 | 27.41 |
| 20% | 0.015 | 0.049 | 0.619 | 0.668 | 24.89 |
| 40% | 0.015 | 0.047 | 0.523 | 0.570 | 22.64 |
| 50% | 0.015 | 0.047 | 0.448 | 0.495 | 21.2 |

Figure 6: Model width distribution after 40% pruning on LLaMA-2.

Figure 7: Mask selection frequency across reasoning tasks and mask candidates.

Table 4: Breakdown of the inference latency of the compressed LLaMA-2-13B model on the A6000 GPU.

Table 5: Training Time Comparison (Hours)

| Method | LLaMA-2-7B | | LLaMA-2-13B | |
|---|---|---|---|---|
| DISP-LLM | 2.41 | | 8.83 | |
| SEAP | Mask | 1.56 | Mask | 2.56 |
| | Router | 4.53 | Router | 8.47 |
| Total | 6.09 | | 11.23 | |

Table 6: Host-device transfer time.

| Model | Size | PCIe 4.0 (64 GB/s) | PCIe 6.0 (256 GB/s) | NVLink (900 GB/s) |
|---|---|---|---|---|
| Dense | 13.5 GB | 0.211 s | 0.053 s | 0.015 s |
| SeAP | 9.45 GB | 0.148 s | 0.037 s | 0.011 s |

learning stage leverages global importance assessment over calibration data to converge quickly, and the router is trained using parameter-efficient LoRA adaptation, adding only $\sim$1% extra parameters. Compared to DISP-LLM, SEAP incurs a $2.53\times$ training-time overhead on LLaMA-2-7B but only $1.27\times$ on LLaMA-2-13B, indicating favorable scaling as the relative cost decreases with model size. The absolute training time stays within practical limits ($<$ 12 hours on 8$\times$A6000), and this one-time cost yields a single dynamic structured model that provides consistent accuracy improvements across many downstream tasks, which static methods cannot achieve.

**Inference Latency Evaluation** Table 4 reports an inference latency breakdown for compressed LLaMA-2-13B on an A6000 GPU (2048 input tokens, 256 generated tokens) in the server-side setting where the backbone, router, and masks all reside on device. The additional overhead introduced by SEAP is minimal: mask assignment takes only 0.015 s and parameter loading 0.047 s, together accounting for less than 1% of total inference time. In contrast, the structured sparsity yields substantial speedups as the compression ratio increases: at 50% sparsity, pre-fill time improves by 36.4% (0.704 s $\to$ 0.448 s), time-to-first-token (TTFT) by 29.7% (0.704 s $\to$ 0.495 s), and decode latency by 22.7% (27.41 s $\to$ 21.20 s). These results show that semantic routing achieves meaningful acceleration while incurring negligible on-GPU overhead.

For xmemory-constrained or edge-like deployments, where the dense backbone is stored in host memory and sparse subnetworks are streamed on demand, it would incur additional host-device transfer time during inference. Table 6 reports theoretical transfer times under typical interconnects. On a PCIe 6.0 $\times$16 link, loading a 30% sparse SEAP subnetwork takes only 0.037 s, compared to 0.053 s for the dense model, while reducing device-resident weights from 13.5 GB to 9.45 GB plus a small 2–3% router/mask overhead. Together, these results validate that SEAP is practical both in high-throughput server settings and in compression-oriented offloading scenarios.

## 6 CONCLUSION

In this paper, we introduced SEAP, a semantic-aware structured pruning framework that performs instance-level dynamic sparsification within a single LLM backbone. SEAP introduces an explainability-guided importance estimator to discover structured pruning patterns from calibration data and employs a lightweight router to assign a subnetwork based on input semantics at the pre-fill stage, adapting sparsity to the input while retaining hardware-friendly dense kernels with small memory and latency overheads. Experiments on LLaMA-2/3, Qwen2, and Phi-2 show that SEAP outperforms SOTA structured pruning baselines on language modeling and a range of reasoning and QA tasks. Future work includes exploring richer router training strategies and data mixtures, extending SEAP to more diverse tasks (e.g., multilingual and generative tasks), and combining it with complementary techniques such as hierarchical masking and quantization.

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

## APPENDIX OUTLINES OF SEAP

This appendix provides extended technical and experimental details that support the findings in the main paper. It is organized as follows:

- Section A: Ethics Statement
- Section B: Reproducibility Statement
- Section C: LLM Usage
- Section D: Quantitative Visualizations
- Section E: Implementation Details
- Section F: Limitations and Future Work

## A  ETHICS STATEMENT

This work adheres to the ICLR Code of Ethics. In this study, no human subjects or animal experimentation was involved. All datasets used, including WikiText-2, Alpaca, ARC-Easy, ARC-Challenge, Winogrande, HellaSwag, PIQA, BoolQ, and OpenBookQA, were sourced in compliance with relevant usage guidelines, ensuring no violation of privacy. We have taken care to avoid any biases or discriminatory outcomes in our research process. No personally identifiable information was used, and no experiments were conducted that could raise privacy or security concerns. We are committed to maintaining transparency and integrity throughout the research process.

## B  REPRODUCIBILITY STATEMENT

We have made every effort to ensure that the results presented in this paper are reproducible. All code has been attached as supplementary material to facilitate replication and verification. The experimental setup, including training steps, model configurations, and hardware details, is described in detail in the paper. We have also provided a full description of our semantic-aware structured pruning framework (SEAP) with explainability-guided importance estimation and router-based mask assignment, to assist others in reproducing our experiments. Additionally, all publicly available datasets used in the paper, such as WikiText-2, ARC-Easy, ARC-Challenge, Winogrande, HellaSwag, PIQA, BoolQ, and OpenBookQA, are publicly available, ensuring consistent and reproducible evaluation results. We believe these measures will enable other researchers to reproduce our work and further advance the field.

## C  LLM USAGE

Large Language Models (LLMs) were used to aid in the writing and polishing of the manuscript. Specifically, we used an LLM to assist in refining the language, improving readability, and ensuring clarity in various sections of the paper. The model helped with tasks such as sentence rephrasing, grammar checking, and enhancing the overall flow of the text. It is important to note that the LLM was not involved in the ideation and research methodology. All research concepts, ideas, and analyses were developed and conducted by the authors. The contributions of the LLM were solely focused on improving the linguistic quality of the paper. We have ensured that the LLM-generated text adheres to ethical guidelines and does not contribute to plagiarism or scientific misconduct.

## D  QUANTITATIVE VISUALIZATIONS

**The impact of Calibration Data for Static Pruning**  Recent works Ji et al. (2025); Williams & Aletras (2024) have shown that the choice of calibration data could significantly affect the performance of static unstructured pruning methods. We perform a similar experiment using a classical structured pruning method, FLAP An et al. (2024), on LLaMA-2-7B. We use the calibration data, including PTB, Wikitext-2, and RedPajama, and find that the similar phenomenon also exists in structured pruning. As shown in Figure 8, the per-task performance from different calibration datasets

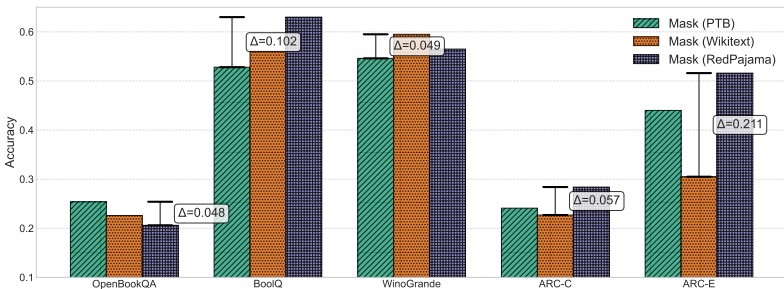

Figure 8: Performance differences of various calibration data using FLAP An et al. (2024) on LLaMA-2-7B.

could vary significantly, showing that the necessity of dynamic pruning. For instance, for ARC-Easy, the performance of FLAP using WikiText-2 could be even dropped by over 50% compared to the performance using RedPajama. This indicates that the choice of calibration data is crucial for structured pruning methods, and dynamic pruning can adaptively select the most suitable calibration data for each task.

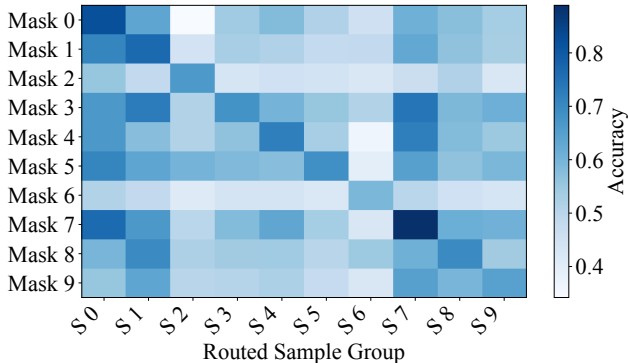

Figure 9: Performance of each mask on each routed group.

**Semantic-aware Sparsity Patterns**    To validate router effectiveness, we evaluate each mask's performance on samples assigned to every mask, creating the router assignment validation matrix shown in Figure 9. The obvious diagonal dominance (0.78 vs. 0.52 off-diagonal performance) confirms the router correctly identifies optimal mask-input pairings, providing compelling evidence that **different instructions benefit from distinct sparsity patterns that preserve semantic-relevant computational pathways.** Importantly, the matrix reveals diverse mask specialization patterns: high-performing specialists like Mask 7 (0.95 peak accuracy, high variance) excel on specific input semantics but fail on others, while moderate-performing generalists like Mask 2 (0.62 average accuracy, low variance) maintain consistent but suboptimal performance across diverse inputs. This performance-specialization trade-off demonstrates that our framework successfully learns a spectrum of pruning strategies, each optimized for different semantics.

**Mask Structural Diversity**    The Hamming distance matrix in Figure 10 reveals substantial structural diversity among learned masks, with distances ranging from 0.18-0.25, indicating masks differ in 18-25% of their pruning decisions. The uniform distance distribution demonstrates that our framework discovers principled complementary patterns rather than converging to similar solutions or generating random variations. This structural diversity, combined with the performance specialization patterns, confirms each mask captures distinct computational pathways optimized for different input semantics.

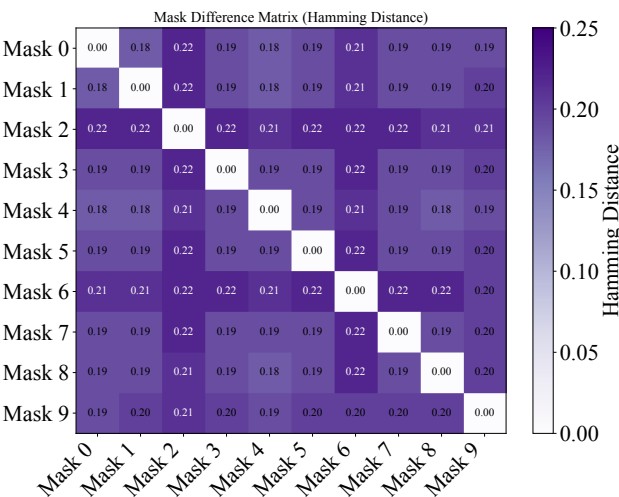

Figure 10: Hamming Distance Between Mask Candidates.

# E    IMPLEMENTATION DETAILS

**Mask Candidate Learning**    We propose an explainability-guided mask candidate learning method based on neuron importance explanation, which provides a principled way to discover diverse pruning patterns optimized for different input characteristics.

**Architecture**    Our dynamic hypernetwork takes layer-wise activations and LRP scores as inputs to generate input-adaptive pruning masks. The architecture consists of:

- **Input Processing:** Layer-wise activations and LRP scores from all transformer layers;
- **Fusion Module:** Lightweight per-layer MLPs that combine activations with attribution-guided importance estimation (Equation 7);
- **Mask Generation:** Binary ReinMax sampling to produce differentiable discrete masks;
- **Temperature Annealing:** Progressive temperature cooling from T_start to T_end during training.

**Hyperparameter Setup**    We train the dynamic hypernetwork using PyTorch with mixed precision (FP16/BF16) training on single GPU. Per-layer MLPs use hidden dimension 128 with scaling factor $\alpha$=1.0 for LRP relevance scores. Temperature annealing progresses from T_start=0.5 to T_end=0.1 during training. For optimization, we use the AdamW optimizer with learning rate 2e-4, weight decay 0.05, and cosine annealing scheduler decaying to minimum learning rate 1e-5. The model is trained for 3 epochs with batch size 1 on unlabeled corpus using perplexity loss.

**Router Learning**    We propose a lightweight semantic-aware router to dynamically assign optimal masks for each input during inference. The architecture consists of:

- **Semantic Encoder:** Pre-trained Embedding model with LoRA fine-tuning;
- **Feature Extraction:** Last token pooling with L2 normalization for robust prompt representations;
- **Prediction Head:** Two-layer MLP classifier that maps semantic embeddings to mask selection;
- **Assignment Strategy:** Argmax selection for deterministic mask assignment during inference.

**Hyperparameter Setup**    We train the router using PyTorch with mixed precision (FP16) training and gradient scaling on single GPU. LoRA adaptation employs rank 8, alpha 32, and dropout 0.05, targeting all linear projection layers. The prediction head uses hidden dimension 256 with ReLU activation and dropout 0.1 for regularization. For optimization, we use the AdamW optimizer with learning rate 1e-5, weight decay 0.01, and cosine annealing scheduler with 10% warmup steps.

The model is trained for 5 epochs with batch size 8 using cross-entropy loss on pre-processed data containing text-mask score pairs. Training data includes samples from five multiple-choice datasets (ARC, PIQA, HellaSwag, Winogrande, and BoolQ) to ensure robust generalization across diverse semantic patterns.

**Baseline Selection**    We compared SEAP against SOTA structured pruning methods, including:

- **FLAP** An et al. (2024) uses fluctuation-based adaptive structured pruning that determines layer importance based on output feature map recoverability and applies compensation mechanisms;
- **SliceGPT** Ashkboos et al. (2024) applies dimensionality reduction by systematically deleting rows and columns from weight matrices using a fixed transformation;
- **ShortGPT** Men et al. (2025) identifies and removes redundant layers by measuring layer importance through input-output cosine similarity;
- **DISP-LLM** Gao et al. (2024) learns a dimension independent fixed pruning matrix on calibration data through global constraint optimization;

We exclude Pudding Wee et al. (2025), a very recent method (within the last two months), because official code is not yet available; we will add it to our evaluation once an implementation is released. For the baseline implementation, we follow the original papers' guidelines and codebases, and evaluate them on the same datasets and metrics as SEAP.

# F    LIMITATIONS AND FUTURE WORK

Our experimental observations reveal an intriguing phenomenon where different input complexities exhibit distinct mask selection patterns, where simple queries can be resolved by multiple masks while challenging problems require specialized ones. This suggests that learned mask archetypes correlate with task difficulty levels, presenting exciting opportunities for future enhancement. We plan to develop difficulty-aware mask learning mechanisms that incorporate uncertainty quantification and confidence-based routing for more adaptive assignment strategies. Additionally, exploring hierarchical mask architectures that dynamically adjust granularity based on task demands could further improve performance across diverse input complexities.

