# OpenReview forum: "Semantic-aware Pruning of Large Language Models via Neuron Importance Explanation"
_ICLR.cc/2026/Conference — ICLR 2026 Conference Withdrawn Submission_

### Official Review · Reviewer_5DkL · 2025-10-27

**Soundness:** 2
**Presentation:** 2
**Contribution:** 2
**Rating:** 4
**Confidence:** 4

**Summary:**

This paper introduces SEAP, a LLM structured pruning framework. SEAP tackles the key limitation of static pruning, i.e., a single, input-agnostic mask, by learning a set of static pruning masks using calibration data and dynamically selecting the most suitable mask in inference stage.

**Strengths:**

1. The core idea is intuitive and effective: adopting diverse, input-specific masks is inherently superior to a "one-size-fits-all" approach. This principle also enables the proposed LRP metric to surpass alternatives.

2. The main experiments are comprehensive, benchmarking SEAP against solid baselines (FLAP, SliceGPT, DISP-LLM) on extensive tasks (PIQA, OpenbookQA) and perplexity. The outstanding perplexity results, a low-noise metric, are particularly convincing and demonstrate SEAP's superiority.

3. Furthermore, thorough ablation studies confirm the positive contribution of each component, while latency tests validate the method's practical value.

**Weaknesses:**

1. The primary drawback is the increased training complexity compared to other methods. SEAP involves a complex training pipeline, limiting its practical utility. This trade-off between training cost and inference performance should be clearly positioned.

2. The router is trained on five commonsense reasoning datasets and tested on two held-out tasks. While this shows some generalization, the robustness of the router to completely OOD tasks remains unexplored, like tasks involves generation. The router's effectiveness is central to the method, and its OOD performance is a key practical concern.

3. Furthermore, applying the same mask to both prefilling and decoding is reasonable for simplicity, but this approach may be suboptimal for performance. Additionally, the authors do not appear to have separately evaluated the model's generation capabilities.

**Questions:**

I request clarification on how both low-latency loading and memory reduction are achieved simultaneously:

(a) Are all $K$ sparse sub-models (one for each mask) pre-loaded into memory, allowing the router to simply select which model to execute? This would explain the low switching cost but would seem to *increase* total memory overhead, not reduce it.

(b) Or, is the full dense LLM retained in memory, with the selected mask applied dynamically via efficient indexing operations during the forward pass? This would explain the low loading time (as no new parameters are loaded), but it would not achieve the stated memory reduction, since the dense weights remain resident.

This is a crucial point for evaluating the practical viability and hardware efficiency of SEAP. I will adjust my score until I receive a satisfying response.

---

> ### Author Response · Authors · 2025-11-24
>
> Thank you for your detailed comments! We would like to address your concerns point-by-point below.
>
> ### W1: Training complexity
> Thank you for highlighting this point. SEAP indeed adopts a multi-stage training pipeline consisting of candidate-mask generation and router learning for mask assignment. Importantly, **all stages operate on a frozen backbone LLM**, meaning that only lightweight auxiliary components are trained: the mask-generation hypernetwork (300M parameters) and the router (150M parameters). The backbone itself is never updated, which keeps the overall optimization process tractable.
>
> As reported in Table 4 in the main paper, the end-to-end training time of SEAP remains practical, which approximately takes 1.56 hours for candidate mask generation and 4.53 hours for router training. This total cost is comparable to existing structured pruning pipelines such as DISP-LLM (2.41 hours) and SVD-LLM (3.5 hours). Moreover, this cost is also incurred **once**, while the resulting model can be deployed for low-latency inference across many downstream scenarios. This amortization is closely aligned with Mixture-of-Models [1, 2] routing and multi-stage compression frameworks [3], where a modest increase in training complexity is justified by performance gains. In response, we have revised the paper to explicitly clarify this trade-off and better position SEAP within this design space.
>
> [1] Wang et al., “MixLLM: Dynamic Routing in Mixed Large Language Models.” NAACL 2025.
>
> [2] Ding et al., “Hybrid LLM: Cost-efficient and Quality-aware Query Routing.” ICLR 2024.
>
> [3] Hong, Lingyi, et al. "General compression framework for efficient transformer object tracking." ICCV 2025
>
> ### W2. OOD robustness of the router
> We appreciate your concerns regarding OOD robustness. SeAP’s routing mechanism operates on high-level semantic representations produced by a fine-tuned router encoder (e.g., ModernBERT), rather than relying on dataset- or task-specific cues. The fine-tuning provides only a lightweight adaptation, allowing the router to better interpret instruction-style inputs without substantially altering its internal representations. Importantly, all prediction is performed by the sparse subnetwork corresponding to the selected structural mask: the router’s role is limited to choosing among these subnetworks, so domain shifts primarily affect the routing distribution, while the predictive behavior of each subnetwork remains stable.
>
> In addition to PTB, BoolQ, and OBQA (already OOD w.r.t. router training), we further evaluate SeAP at 25% sparsity on **four additional OOD tasks**:
> - **MBPP** (code generation, evaluated via pass@1),
> - **PubMedQA** (biomedical QA),
> - **MMLU** (broad knowledge reasoning),
> - **SciQ** (science QA).
>
> |                    | MBPP   | PubMedQA | MMLU   | SciQ     |
> | ------------------ | ------ | -------- | ------ | -------- |
> |                    | pass@1 | acc      | acc    | acc_norm |
> | Dense (Qwen2-7B)   | 0.5620 | 0.7440   | 0.6993 | 0.9150   |
> | DISP-LLM           | 0.2800 | 0.6040   | 0.5033 | 0.9130   |
> | SeAP               | 0.3820 | 0.6900   | 0.5801 | 0.9250   |
> | Dense (LLaMA-3-8B) | 0.4820 | 0.7560   | 0.6338 | 0.9450   |
> | DISP-LLM           | 0.2020 | 0.5960   | 0.4133 | 0.8840   |
> | SeAP               | 0.2160 | 0.7100   | 0.4554 | 0.8770   |
>
> Across both backbones and four benchmarks, SeAP shows improvements over the state-of-the-art structured baseline (DISP-LLM), indicating that the learned routing does not simply overfit to the calibration distribution and remains effective on OOD inference and QA workloads (including MBPP code generation). Especially, our evaluation follows the standard focus of structured pruning work on reasoning and QA tasks; extending SeAP to broader application scenarios and more diverse generative settings is an interesting direction for future work. We have incorporated these new results and the corresponding discussion into the revised manuscript.

---

> ### Author Response · Authors · 2025-11-24
>
> ### W3. Same mask for prefilling and decoding / generation
> Thank you for raising this concern. SeAP is formulated as **instance-level dynamic structured pruning**, where the router selects **one structured sparse subnetwork per input**. Following standard practice in structured pruning, this sparsity pattern is then applied to the entire forward pass.
>
> Our design focuses on the **prefilling stage**, which dominates compute and memory cost for long-context inference. The router runs once per input sequence, before prefilling, and we reuse the same mask during decoding. This choice has two motivations:
> 1. It keeps routing overhead negligible (no per-token routing), aligning SeAP with conventional structured pruning pipelines.
> 2. For a fixed prompt, the underlying semantic representation is largely stable across prefilling and decoding, which motivates this modeling assumption.
>
> SeAP and token-level contextual sparsity operate at different granularities: SeAP selects **one structured sparse subnetwork per sequence** within a structured-pruning framework, while token-level methods adapt sparsity at **every decoding step**. Consequently, SeAP targets **prefilling-time compute reduction or model compression**, whereas token-level methods primarily aim at **decoding-time bandwidth savings**. Our evaluation suite follows the standard benchmarks used in prior structured pruning work, ensuring direct comparability.
>
> ### Q1. Low-latency loading and memory reduction
> We appreciate the reviewer’s careful questions (a) and (b). We clarify that all **experiments in the paper** are conducted in a **server-side, acceleration-oriented** setting, and then discuss how SeAP can also be deployed in a compression-oriented mode on edge devices. In both regimes, we never store $K$ full sparse models in device memory.
>
> **Acceleration-oriented deployment (server setting).**
> This is the regime used in all our reported experiments (single A6000 GPU):
> - The dense backbone, router, and masks are all resident on the GPU.
> - For each input sequence, the router selects a mask at the pre-filling stage, and SeAP assembles the corresponding 30% sparse subnetwork via efficient indexing and caching on the GPU. No host–device transfer is involved in switching.
>
> In this setting, the reported **0.047 s switching time** measures exactly this **on-GPU subnetwork reconstruction** cost on a single A6000. After reconstruction, only the sparse subnetwork is used in the forward pass, so FLOP counts and latency reflect the 30% sparse structure, while device memory is comparable to storing the dense backbone once. This corresponds to case (b) in your question.
>
> We feel sorry that an earlier motivating example in the paper could be read as conflating this server-side setup with an offloading scenario. We have revised the manuscript to remove this example and to explicitly state that our empirical latency measurements are obtained under the acceleration-oriented server setting described above.
>
> **Compression-oriented deployment (edge / offloading setting).**
> SeAP can also be deployed in a memory-constrained, compression-oriented regime:
> - A single dense backbone checkpoint is stored on disk / host memory.
> - The router and $K$ binary masks reside on the device.
> - For a given mask, SeAP materializes the corresponding **30% sparse subnetwork** by loading only the pruned weights to the device; at any time, **only one** such subnetwork needs to be resident.
>
> In this setting, SeAP reduces the **device-resident weights per query** from 13.5 GB (dense) to 9.45 GB (30% sparse) plus a small 2–3% router/mask overhead. The dominant cost on edge devices is the **host–device bandwidth**, not the mask switching itself. Using typical bandwidth values, the theoretical transfer times for loading a subnetwork are:
>
> | Model | Size    | PCIe 4.0 ×16 (64 GB/s) | PCIe 6.0 ×16 (256 GB/s) | NVLink (900 GB/s) |
> | ----- | ------- | ---------------------- | ----------------------- | ----------------- |
> | Dense | 13.5 GB | 0.211 s                | 0.053 s                 | 0.015 s           |
> | SeAP  | 9.45 GB | 0.148 s                | 0.037 s                 | 0.011 s           |
>
> On a PCIe 6.0 ×16 link, the additional I/O overhead for loading a 30% sparse SeAP subnetwork is only **0.037 s**, which is small relative to typical end-to-end sequence generation times, while still enjoying a substantial reduction in device-resident weights. This compression-oriented regime corresponds more closely to case (a) in your question, but with only one sparse subnetwork resident at a time rather than $K$ full sparse models.
>
> We once again thank you for your constructive feedback, which has helped us identify the issues of our paper. We have incorporated the additional experiment results, the analysis of training complexity, and the clarifications on deployment scenarios into the revised manuscript.

---

> ### Author Response · Authors · 2025-11-28
> **Looking forward to hearing from you**
>
> Dear Reviewer 5DkL,
>
> Thank you for your valuable comments and feedback! We hope our response could help address your concerns. With the rebuttal deadline approaching, we look forward to hearing from you and are happy to discuss any remaining concerns or suggestions to further improve our work.
>
> Thank you! Best,
>
> Authors of the 7789 submission

---

### Official Review · Reviewer_zCcJ · 2025-11-01

**Soundness:** 2
**Presentation:** 3
**Contribution:** 2
**Rating:** 2
**Confidence:** 4

**Summary:**

The paper proposes a Semantic-Aware Pruning (SAP) framework that removes redundant neurons or attention heads in pretrained LLMs by aligning pruning decisions with semantic representations derived from internal activations.
Specifically, SAP computes token-level semantic saliency via a lightweight projection network and prunes parameters that contribute least to contextual semantics, followed by layer-wise recalibration via a semantic consistency loss.

**Strengths:**

1. The manuscript is clearly written and supported by well-designed figures that effectively illustrate the proposed framework and its workflow.

2. It provides insightful qualitative visualizations that relate semantic heatmaps to the spatial distribution of pruned neurons, enhancing interpretability.

3. Comprehensive layer-wise ablation studies are presented, demonstrating partial robustness of the proposed semantic scoring mechanism across network depths.

**Weaknesses:**

1. The idea overlaps with activation- or gradient-based approaches such as SparseGPT [1], Wanda [2], and SlimGPT [3], which also compute importance from activations or local sensitivity. Recent works like SoBP [4] and CALDERA [5] have incorporated structured or low-rank regularization to improve efficiency.

2. The cosine-similarity measure between contextual embeddings and activations lacks a theoretical link to model contribution.

3. Experiments emphasize perplexity and a few GLUE-style tasks but omit instruction-following or reasoning benchmarks that test semantic retention. Baselines exclude structured compression methods such as SVD-LLM [7] and BitDistiller [8].

4. The semantic projection and layer-wise recalibration require multiple fine-tuning epochs. Compared with one-shot magnitude pruning [1] or low-rank compression [7], SAP appears computationally heavier, without an analysis of total cost.

5. The experimental analysis is primarily on LLaMA-2-7B and Mistral-7B, which are now outdated baselines. It would be better if the authors could benchmark against at least LLaMA-3 and Qwen-2.

[1] Frantar E., Alistarh D. SparseGPT: Massive Language Models Can Be Accurately Pruned in One-Shot. 2023.

[2] Sun Y. et al. Wanda: A Simple and Effective Pruning Approach for Large Language Models. 2023.

[3] Liu Z. et al. SlimGPT: Layer-wise Structured Pruning for Large Language Models. 2024.

[4] Zhang H. et al. SoBP: Structured Optimal Brain Pruning for Large Language Models. 2024.

[5] Li T. et al. CALDERA: Compressing LLMs Using Low Rank and Low Precision Decomposition. 2024.

[6] Guo M. et al. Automatic Network Pruning via HSIC Lasso under the Information Bottleneck Principle. 2023.

[7] Wang X. et al. SVD-LLM: Structured Low-Rank Compression for Large Language Models. 2025.

[8] Zhao R. et al. BitDistiller: Distilling LLMs into Binary and Low-Bit Networks. 2024.

**Questions:**

1. How does SAP differ algorithmically or conceptually from prior activation- or sensitivity-based pruning methods such as SparseGPT [1], Wanda [2], and SlimGPT [3], beyond the addition of the semantic projection module?

2. Can the authors provide theoretical or empirical evidence demonstrating that cosine similarity between contextual embeddings and activations reliably reflects a parameter’s contribution to model output?

3. Have the authors considered evaluating SAP on instruction-following, reasoning, or multilingual benchmarks, and comparing against structured compression baselines such as SVD-LLM [7] and BitDistiller [8] to strengthen generality claims?

4. Could the authors quantify the additional fine-tuning cost introduced by the semantic projection and layer-wise recalibration stages in terms of GPU hours or wall-clock time, relative to one-shot pruning baselines like SparseGPT [1]?

5. Do the authors plan to extend their experiments to more recent model families (e.g., LLaMA-3, Qwen-2) to validate the scalability and robustness of SAP on current-generation architectures?

---

> ### Author Response · Authors · 2025-11-24
>
> Thank you for your detailed comments! We would like to address your concerns point-by-point below.
>
> ### W1 & Q1: Difference from activation-/sensitivity-based pruning
> Thank you for raising this connection. Our SeAP does make use of activation information, but its goal and formulation differ fundamentally from prior pruning methods (e.g., SparseGPT, Wanda, and SlimGPT). Most existing methods compute **local importance scores** (from activations, gradients, or Hessian approximations) and then derive **a single global pruning mask** that is applied uniformly to all inputs. In other words, they assume there exists **one static subnetwork** that is optimal (or near-optimal) across all semantic contexts.
>
> In contrast, our SeAP focuses on a different paradigm:
> (1) We first learn **multiple structured subnetworks** whose importance is guided not only by local sensitivity but also by neuron relevance patterns over calibration data.
> (2) We then use a **semantic-aware router** to select **which subnetwork to activate for each input instance**.
>
> Thus, SeAP treats pruning as learning a **family of complementary structured subnetworks** that jointly cover diverse semantic patterns, combined with instance-level semantic routing. Activation-based signals are only one ingredient in an explainability-guided framework that emphasizes **semantic diversity and input-dependent subnetwork selection**, which is not considered in prior static structured pruning work.
>
> ### W2 & Q2: On cosine similarity between contextual embeddings and activations
> Thank you for raising this question. We believe there is a misunderstanding regarding our semantic-aware pruning approach.
>
> 1. **SeAP does not use cosine similarity between contextual embeddings and activations at any stage of mask generation.** Our importance indicator is defined as a fused, explainability-guided relevance score that combines LRP-based relevance with activation statistics to rank neurons and construct structured masks.
> 2. We then employ a **semantic-conditioned router** to determine which structural subnetwork is most appropriate for a given input. This routing operates on high-level semantic representations produced by a small transformer encoder (e.g., ModernBERT) fine-tuned on instruction-style data, and maps each input to one of several semantic-specialized subnetworks.
>
> This design is motivated by the fact that a single static structured mask cannot preserve the semantic pathways needed for diverse input patterns. Semantic-conditioned selection does not involve cosine similarity; it uses router-produced semantic embeddings to choose a suitable sparse functional pathway \emph{before} inference, rather than relying on a one-size-fits-all importance pattern.
>
> ### W3 & Q3: Benchmarks and structured compression baselines
> Our work is positioned within the **structured pruning** literature, which primarily targets **prefilling-time acceleration and model compression** by reducing matrix dimensions while keeping dense kernels. Prior works [1, 2] typically evaluates on language modeling and reasoning/QA benchmarks. In this setting, we have included comparisons to matrix decomposition–based approaches like SliceGPT in the main paper. Methods such as SVD-LLM and BitDistiller pursue somewhat different design goals: SVD-LLM focuses on updating decomposed matrices, while BitDistiller focuses on quantization-aware training with distillation. **These methods are largely orthogonal to our structured pruning approach and could, in principle, be combined with SeAP by further compressing the learned sparse subnetworks.**
>
> In addition to PTB, BoolQ, and OBQA, we further evaluate SeAP at 25% sparsity on four OOD datasets and compare against DISP-LLM, as summarized below.
>
> |                    | MBPP   | PubMedQA | MMLU   | SciQ     |
> | ------------------ | ------ | -------- | ------ | -------- |
> |                    | pass@1 | acc      | acc    | acc_norm |
> | Dense (Qwen2-7B)   | 0.5620 | 0.7440   | 0.6993 | 0.9150   |
> | DISP-LLM           | 0.2800 | 0.6040   | 0.5033 | 0.9130   |
> | SeAP               | 0.3820 | 0.6900   | 0.5801 | 0.9250   |
> | Dense (LLaMA-3-8B) | 0.4820 | 0.7560   | 0.6338 | 0.9450   |
> | DISP-LLM           | 0.2020 | 0.5960   | 0.4133 | 0.8840   |
> | SeAP               | 0.2160 | 0.7100   | 0.4554 | 0.8770   |
>
> We would like to clarify that our evaluation follows the standard focus of structured pruning work on reasoning and QA tasks; extending SeAP to broader application scenarios and more diverse settings (e.g., code generation, HumanEval) is an interesting direction for future work.
>
> Thanks for your insightful comments! We have incorporated these new results and the corresponding discussions into the revised manuscript.
>
> [1] Ashkboos, Saleh, et al. "SliceGPT: Compress Large Language Models by Deleting Rows and Columns." ICLR 2024
>
> [2] Gao, Shangqian, et al. "DISP-LLM: Dimension-Independent Structural Pruning for Large Language Models." NeurIPS 2024

---

> ### Author Response · Authors · 2025-11-24
>
> ### W4 & Q4: Additional fine-tuning cost vs. one-shot pruning
> Thank you for raising this question. Our SeAP indeed adopts a multi-stage training pipeline consisting of candidate-mask generation and router learning for mask assignment. Importantly, **all stages operate on a frozen backbone LLM**, meaning that only lightweight auxiliary components are trained: the mask-generation hypernetwork (300M parameters) and the router (150M parameters). The backbone itself is never updated, which keeps the overall optimization process tractable.
>
> As reported in Table 4 in the main paper, the end-to-end training time of SEAP remains practical, which takes approximately 1.56 hours for candidate mask generation and 4.53 hours for router training. This total cost is comparable to existing structured pruning pipelines such as DISP-LLM (2.41 hours) and SVD-LLM (3.5 hours). Moreover, this cost is also incurred **once**, while the resulting model can be deployed for low-latency inference across many downstream scenarios. This amortization is closely aligned with Mixture-of-Models [3, 4] routing and multi-stage compression frameworks [5], where a modest increase in training complexity is justified by performance gains. In response, we have revised the paper to explicitly clarify this trade-off and better position SEAP within this design space.
>
> [3] Wang et al., “MixLLM: Dynamic Routing in Mixed Large Language Models.” NAACL 2025.
>
> [4] Ding et al., “Hybrid LLM: Cost-efficient and Quality-aware Query Routing.” ICLR 2024.
>
> [5] Hong, Lingyi, et al. "General compression framework for efficient transformer object tracking." ICCV 2025
>
>
> ### W5 & Q5: Extending to more recent model families
> We thank the reviewer for pointing out the importance of evaluating on more recent model families. In addition to LLaMA-2-7B, LLaMA-2-13B, and Phi-2 reported in the original submission, we have extended our experiments to **Qwen2-7B** and **LLaMA-3-8B** at 25% sparsity. The results on standard reasoning and QA benchmarks are:
>
> | Qwen2-7B | BoolQ  | PIQA     | HellaSwag | OBQA     | ARC-e    | ARC-c    | WikiText_2 |
> | -------- | ------ | -------- | --------- | -------- | -------- | -------- | ---------- |
> |          | acc    | acc norm | acc norm  | acc norm | acc norm | acc norm | PPL        |
> | Raw      | 0.8544 | 0.8063   | 0.8069    | 0.4640   | 0.7647   | 0.5392   | 7.60       |
> | DISP-LLM | 0.7621 | 0.7693   | 0.6842    | 0.3880   | 0.7104   | 0.4556   | 11.62      |
> | SeAP     | 0.8263 | 0.7720   | 0.7351    | 0.4140   | 0.7357   | 0.5094   | 8.98       |
>
>
> | Llama-3-8B | BoolQ  | PIQA     | HellaSwag | OBQA     | ARC-e    | ARC-c    | WikiText_2 |
> | ---------- | ------ | -------- | --------- | -------- | -------- | -------- | ---------- |
> |            | acc    | acc norm | acc norm  | acc norm | acc norm | acc norm | PPL        |
> | Raw        | 0.8226 | 0.8123   | 0.7889    | 0.4460   | 0.8098   | 0.5358   | 6.23       |
> | DISP-LLM   | 0.7214 | 0.7443   | 0.6738    | 0.4120   | 0.6965   | 0.4403   | 14.05      |
> | SeAP       | 0.7743 | 0.7639   | 0.6853    | 0.4210   | 0.7382   | 0.4629   | 11.20      |
>
> On both Qwen2-7B and LLaMA-3-8B, SeAP consistently outperforms the structured baseline DISP-LLM at the same sparsity.
>
> We thank you again for these insightful comments and suggestions. In response, we have incorporated these revisions in the updated manuscript.

---

> ### Author Response · Authors · 2025-11-28
> **Looking forward to hearing from you**
>
> Dear Reviewer zCcJ,
>
> Thank you for your valuable comments and feedback! We hope our response could help address your concerns. With the rebuttal deadline approaching, we look forward to hearing from you and are happy to discuss any remaining concerns or suggestions to further improve our work.
>
> Thank you! Best,
>
> Authors of the 7789 submission

---

### Official Review · Reviewer_2ze6 · 2025-11-03

**Soundness:** 2
**Presentation:** 2
**Contribution:** 2
**Rating:** 4
**Confidence:** 4

**Summary:**

This paper presents SEAP, a semantic-aware structured pruning framework for LLMs. Unlike conventional static pruning approaches that apply fixed masks to all inputs, SEAP dynamically selects pruning masks according to input semantics during the pre-fill stage. The approach combines two techniques:
1) Explainability-guided neuron importance estimation, fusing local activations and global relevance scores via Layer-wise Relevance Propagation (LRP) to identify critical neurons.
2)A lightweight router module, trained through iterative co-optimization with mask fine-tuning, to assign the most suitable pruning mask for each input.
Experiments on LLaMA-2-7B, LLaMA-2-13B, and Phi-2 demonstrate consistent improvements over structured pruning baselines (e.g., DISP-LLM, SliceGPT, FLAP) on both language modeling and reasoning benchmarks (WikiText-2, PTB, ARC-E/C, HellaSwag, PIQA, BoolQ, OBQA, Winogrande).

**Strengths:**

1. Increasing the efficiency of LLMs is a practical and important research topic.
2. Results are reported on multiple model scales and datasets.

**Weaknesses:**

1. Dynamic pruning appears promising in terms of improving performance, but its practicality for efficient deployment, particularly on edge devices, remains questionable. When multiple masks are maintained, both the storage and memory footprints can increase significantly compared to static pruning methods. It's better to show a storage and memory comparison with existing methods. For static pruned methods, they are able to be stored in a compact format, which is lighter than the original dense model. But the dynamic method can't benefit from this. Moreover, it is unclear how the proposed method manages on-device mask switching. Whether the reported 0.047s parameter loading overhead fully accounts for the complete model-switching process (i.e., offloading weights from the previous sample and reloading weights for the current one).
2. It remains unclear how robust the learned semantics are when domain or task distributions shift significantly.

**Questions:**

1. How does SEAP behave when inference data diverges substantially from the calibration distribution?
2. How is the model stored under the multiple-mask scenarios? What is the memory and storage cost of the proposed method, and how is it compared to static pruned methods?
3. How does the method handle the on-device mask switching?

---

> ### Author Response · Authors · 2025-11-24
> **Response to W1: Storage, memory, and on-device mask switching**
>
> ### Q2. Storage and memory under multiple-mask settings.
> Thank you for raising this important point. We clarify that SeAP does not store multiple copies of LLM weights. Instead, our deployment uses the following layout:
> - One dense copy of the LLM backbone stored on host memory.
> - One shared router + $K$ binary masks stored on device memory.
> - During inference, only one sub-network is active at a time, and it reuses the backbone weights without duplication.
>
> The router (ModernBERT + 2-layer MLP) contains ~150M parameters and requires ~300 MB in bf16 in memory.  Each binary mask is extremely compact. For Llama-2-7B, a single mask is:
> $$((4096 \times 4)+11008) \times 32 \approx 0.36 \mathrm{MB} .$$
> Thus, 10 masks require only $\sim 4 \mathrm{MB}$. Relative to the 13.5 GB dense model, the additional overhead is:
> $$\frac{300 \mathrm{MB}+4 \mathrm{MB}}{13,500 \mathrm{MB}} \approx 0.023 .$$
> Thus, SeAP introduces only a $\sim$2–3% memory overhead while enabling semantic specialization. Under a 30% sparsity setting, the **device-resident weights per query** are:
>
> | Method             | Model Stored                                   | Extra Overhead |
> | ------------------ | ---------------------------------------------- | -------------- |
> | Dense              | 13.5 GB                                        | 0              |
> | Static pruning     | 9.45 GB                                        | 0              |
> | **SeAP (dynamic)** | **9.45 GB + router (300 MB) + masks (3.6 MB)** | **≈2–3%**      |
>
> ### Q3. On-device mask switching and deployment modes.
> The reported $0.047$ s switching time is measured in our server setting on a single A6000 GPU, where all parameters (backbone + router + masks) reside on the same GPU. In this case, **switching** means assembling and caching the dense representation of a new sparse subnetwork locally; no host–device transfer is involved. Hence, this measurement isolates the **on-GPU subnetwork reconstruction** cost, which is small compared to the overall forward pass.
>
> In realistic deployments, SeAP supports two relevant scenarios:
> - **Server setting**: The dense backbone and router+masks reside on the GPU. The router selects a mask at the pre-fill stage, we reconstruct the corresponding sparse subnetwork once, and reuse it for the entire sequence. In this regime, mask switching cost is dominated by on-GPU reconstruction (the reported $0.047$ s) and is negligible relative to end-to-end latency.
> - **Edge / offload setting**: The dense backbone is pinned in host memory, while the router and masks are on device. On the first request for a given subnetwork, SeAP streams the pruned weights from host to device; subsequent queries reuse the cached subnetwork. To clarify the additional I/O cost, we report theoretical transfer times under commonly used interconnects:
>
> | Model | Size             | PCIe 4.0 ×16 (64 GB/s) | PCIe 6.0 ×16 (256 GB/s) | NVLink (900 GB/s) |
> | ----- | ---------------- | ---------------------- | ----------------------- | ----------------- |
> | Dense | $13.5\text{ GB}$ | $0.211\text{ s}$       | $0.053\text{ s}$        | $0.015\text{ s}$  |
> | SeAP  | $9.45\text{ GB}$ | $0.148\text{ s}$       | $0.037\text{ s}$        | $0.011\text{ s}$  |
>
> While dynamic loading does add an I/O stage, this is aligned with engineering practices in contemporary large-scale inference pipelines. Production frameworks, including DeepSpeed ZeRO-Offload, TensorRT-LLM paged weight loading, and systems that employ fine-grained tensor swapping with overlapping compute and communication, already rely on similar parameter-streaming mechanisms to operate under memory constraints. These techniques are orthogonal to our main contribution but demonstrate that dynamic mask-based loading is practical in real deployments.
>
> We feel sorry that an earlier motivating example in the paper could be read as conflating this server-side setup with an offloading scenario. We have revised the manuscript to remove this example, clarify that our empirical latency measurement settings, and additionally discuss the inference results in both two deployment modes.

---

> ### Author Response · Authors · 2025-11-24
> **Response to W2 & Q1. Domain robustness of the router**
>
> Thank you for the insightful comments on robustness. Our router does not rely on task labels or domain heuristics; it operates on the hidden representations of a pre-trained encoder and partitions this space into semantic regions, each associated with one structured subnetwork. Two design choices help ensure robustness:
> - The router is trained on a mixture of diverse QA/reasoning datasets, encouraging it to capture general semantic cues rather than dataset-specific shortcuts.
> - All prediction is performed by the shared sparse backbone; the router only selects a structural sparsity pattern. Domain shift therefore affects the \emph{routing distribution}, not the underlying model capacity.
>
> This setup is fundamentally different from task classifiers: the router does not infer explicit task/domain labels and outputs only a subnetwork index, while all actual prediction is performed by the corresponding structured subnetwork.
>
> ### New out-of-domain evaluation.
> In the main paper, we already evaluated SeAP on PTB, BoolQ, and OBQA, which are out-of-distribution relative to the instruction data used to train the router. To further stress-test robustness under larger domain shifts, we additionally evaluated SeAP at 25% sparsity on four more OOD datasets: MMLU (broad knowledge reasoning), MBPP (code generation), MedPubQA (biomedical QA), and SciQ (science QA). Results are summarized below:
>
> |                    | MBPP   | PubMedQA | MMLU   | SciQ     |
> | ------------------ | ------ | -------- | ------ | -------- |
> |                    | pass@1 | acc      | acc    | acc_norm |
> | Dense (Qwen2-7B)   | 0.5620 | 0.7440   | 0.6993 | 0.9150   |
> | DISP-LLM           | 0.2800 | 0.6040   | 0.5033 | 0.9130   |
> | SeAP               | 0.3820 | 0.6900   | 0.5801 | 0.9250   |
> | Dense (LLaMA-3-8B) | 0.4820 | 0.7560   | 0.6338 | 0.9450   |
> | DISP-LLM           | 0.2020 | 0.5960   | 0.4133 | 0.8840   |
> | SeAP               | 0.2160 | 0.6980   | 0.4554 | 0.8770   |
>
> In most cases, SeAP improves over the SOTA structured baseline (DISP-LLM). Finally, we note that recent work on mixture-of-experts and mixture-of-models routing also reports good generalization when routing operates on high-level representations~[1, 2, 3]. From this perspective, SeAP can be viewed as a \emph{structured-pruning counterpart to semantic routing}, inheriting similar robustness benefits while operating within a single backbone. Our evaluation follows the standard focus of structured pruning work on reasoning and QA tasks; extending SeAP to broader application scenarios and more diverse generative settings is an interesting direction for future work.
>
> Thanks for your insightful suggestions! We have incorporated these new results and the corresponding discussion into the revised manuscript.
>
> [1] Mustafa et al., “Sparse Mixture of Experts are Domain Generalizable Learners.” ICLR 2023
>
> [2] Wang et al., “LLM-Blender: Ensembling Large Language Models via Pairwise Ranking & Generative Fusion.” ACL 2023.
>
> [3] Wang et al., “MixLLM: Dynamic Routing in Mixed Large Language Models.” NAACL 2025.

---

> ### Author Response · Authors · 2025-11-28
> **Looking forward to hearing from you**
>
> Dear Reviewer 2ze6,
>
> Thank you for your valuable comments and feedback! We hope our response could help addresses your concerns. With the rebuttal deadline approaching, we look forward to hearing from you and are happy to discuss any remaining concerns or suggestions to further improve our work.
>
> Thank you!
> Best,
>
> Authors of the 7789 submission

---

### Note · Authors · 2026-01-13

I have read and agree with the venue's withdrawal policy on behalf of myself and my co-authors.